materials science

lead-acid battery, scrap lead paste, tetrabasic lead sulfate, crystal seed, cycle life

**Author for correspondence:**
Myonghak Kim
e-mail: kmh311@163.com

# Study on synthesis and application of tetrabasic lead sulfate as the positive active material additive for lead-acid batteries

Myonghak Kim[1,2], Mungi Kim[1], Cholnam Ri[3], Songchol Jong[1], Ilman Pak[1], Ganghyok Kim[1] and Mun Ri[2]

[1]Faculty of Applied Chemical Engineering, Kim Chaek University of Technology, No. 60 Pyongyang Kyogu, Democratic People's Republic of Korea
[2]School of Chemical Engineering, China University of Petroleum, No. 66 Changjiang West Road, Qingdao 266580, People's Republic of China
[3]Institute for Electronic Materials, Kim Il Sung University, Pyongyang, Democratic People's Republic of Korea

MK, 0000-0001-6619-5670

Tetrabasic lead sulfate (4BS) was used as a positive active material additive for lead-acid batteries, which affirmatively affected the performance of the battery. Herein, tetrabasic lead sulfate was synthesized from scrap lead paste that was formed through the production process of the lead-acid batteries. This solves the disposing problem of the scrap lead paste that is challenging in the production of the lead-acid batteries. Scrap lead paste was first pre-treated and the 4BS with high purity and crystalline was synthesized by sintering at the temperature of 450°C and hold time of 7 h. As demonstrated by X-ray diffraction and scanning electron microscopy test and Material Studio software calculation, the purity of synthesized 4BS is higher than 98 wt%, small particles have pillar forms and are evenly distributed. Moreover, the synthesized 4BS of 1 wt% was added to the positive lead paste and then valve-regulated lead-acid battery was made after the pasting, curing and formation processes. The effectiveness of the lead-acid batteries after adding 4BS as crystal seeds was evaluated, and the 100% charge–discharge cycle life of the new battery (523 times) was about 1.4 times higher than that of general lead-acid batteries (365 times).

# 1. Introduction

Lead-acid batteries are widely used in various applications because of their advantages like high efficiency, low cost, security and stable performances [1]. Therefore, lead-acid batteries are the most versatile and reliable chemical power source for practical applications. In particular, lead-acid batteries have been extensively used in electric bicycles, hybrid or electrical vehicles, energy storage [2] and many other applications over the last few decades. They account for about 50% of the battery market [3].

However, at present, the development of lead-acid batteries needs its higher specific capacity, higher specific power, and longer cycle life for hybrid electrical vehicle or electrical vehicle industry [4–7]. The present performances of lead-acid batteries are not adequate for various emerging applications. Therefore, research attention has been focused on improving the specific capacity, specific power, charge acceptance, rate capability and cycle life of the valve-regulated lead-acid (VRLA) battery [8]. The most common damage mechanisms for a VRLA battery include positive grid alloy corrosion, positive active material softening and shedding, irreversible sulfating in the negative electrode, water loss, electrolyte stratification, internal short circuit and so on [9,10]. In these damage mechanisms, it is believed that the positive electrode softening and shedding is one of the essential factors and is the main reason for the failure of the positive plate. Actually, research of VRLA battery's service life is mainly to reduce these effects. The positive paste of the battery is congregated from a mixture of Pb, PbO and $Pb_3O_4$, and during the curing and formation processes, the transformation of the material is responsible for the structural integrity and electrical contact among the active material particles.

From the positive plate manufacturing process, curing is a key stage and then positive electrode material is transformed into tribasic lead sulfate (3BS) and tetrabasic lead sulfate (4BS). The relative amounts of 3BS and 4BS in the positive paste are influenced by the proprieties of starting leady oxide, the conditions of the curing process, positive active material additive and so on. Moreover, the constituent of positive active material transforms into two kinds of lead dioxide in the process of formation. Tribasic lead sulfate converts to β-$PbO_2$ and the tetrabasic lead sulfate converts to α-$PbO_2$. The crystal structure of these two kinds of lead dioxide is substantially different; β-$PbO_2$ has higher discharge capacity due to its better electrochemical activity than α-$PbO_2$. However, α-$PbO_2$ has the larger crystal size and harder particles, so it can be the positive active material skeleton in order to increase the cycle life of the lead-acid batteries. 3BS crystals decide the high capacity of positive active material, and 4BS decides its good cycle life. Changing the ratio of 3BS and 4BS can directly affect the capacity and cycle life of the positive plate (lead dioxide).

Many researchers also found that tetrabasic lead sulfate can be a type of ideal positive active material additive to promote the generation of 4BS in the lead paste during the curing process [11–13]. Adding it can prevent lead-acid batteries from the early capacity loss due to the effect of loss of active materials and result in the significant increase in cycle life. 4BS has such big crystal grains that they form network skeletons in positive active materials, and using 4BS prevents the positive active materials from softening and shedding. Therefore, the life of the battery is increased and especially the overcharge characteristic of the battery is improved.

In the early days, many researchers [10] tried to get the increased amount of 4BS in the positive paste by the high-temperature paste manufacture or the high-temperature curing process, so that the mechanical strength of the positive plate could become higher and the cycle life could be lengthened [14,15]. Through the studies of Pavlov et al. [16,17], they considered that the mixing lead paste temperature can determine the proportion of 3BS and 4BS in the positive lead paste. And they all found that the proportion of 4BS in the lead paste was increased with the rising of temperature. The more 4BS is produced on the positive plate after curing and the obtained 4BS particles are coarser if curing time is longer and the temperature is higher. Such 4BS is useful to increase the cycle life of lead-acid batteries. Additionally, the initial capacity of the battery became to decrease and the formation of the plate became to be difficult, and the combined operation of batteries had a problem because of coarser 4BS.

Thus, it is difficult to control the content and crystal size of 4BS that was produced through curing process at high temperature. In order to solve this problem, crystal seed is added to limit the growth of 4BS crystal in the curing process and control the size of crystal particles [18]. The seed crystal is just 4BS that was prepared outside and added. Through the studies of Boden et al. [19,20], they considered that electrochemical performance of the lead-acid batteries was improved by adding the synthesized 4BS of 1 wt% in the positive active material.

Several methods including thermal decomposition of 3BS [21,22], hydrothermal method [23,24], Penox method [25] and mechanical grinding [26] are used for the preparation of 4BS but they have a

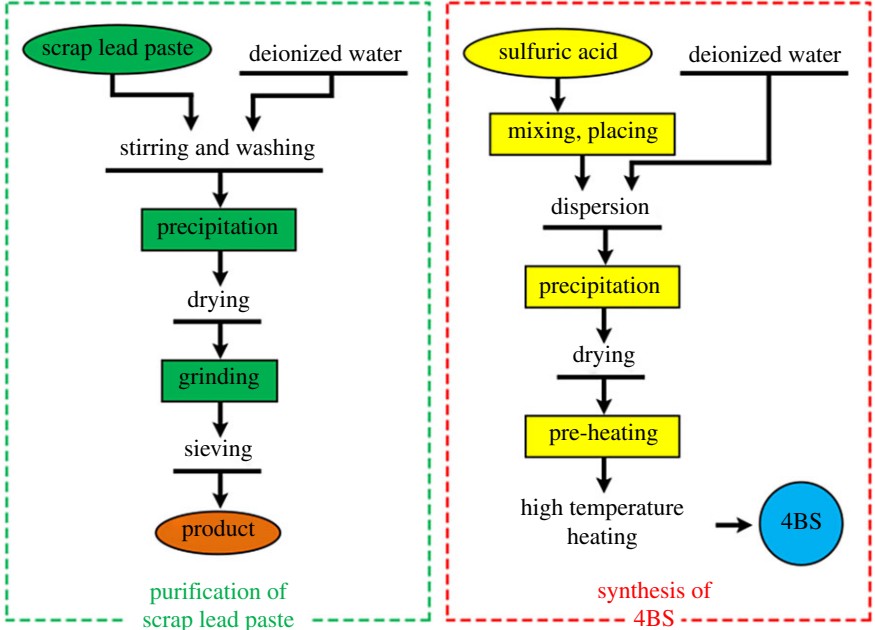

**Figure 1.** Processing diagram for preparation of 4BS from scrap lead paste.

requirement for the high purity of the raw materials and its cost is too much. Meanwhile, scrap lead paste is inevitably produced through pasting in the production of lead-acid batteries [27,28]. If it is used for the next period of production in pasting, the performance of batteries is decreased. And it has difficulty with being recycled and is thus considerable waste, which is harmful to the environment.

In this paper, the aim of our work is the synthesizing of 4BS using scrap lead paste from the production of lead-acid batteries and the addition of 4BS to the positive active material of VRLA. It is to improve its electrochemical performance and solve the waste during the production of lead-acid batteries.

# 2. Experimental

## 2.1. Preparation of 4BS from scrap lead paste

All chemicals reagents used in the experiment were purchased from Aladdin Chemical Co., Ltd and used without further purification.

Figure 1 shows the processing diagram for preparing 4BS from scrap lead paste. Preparation of 4BS mainly has two processes: the purification of scrap lead paste and the synthesis of 4BS.

First, the purification of the scrap lead paste was carried out according to the following procedure:

The scrap lead paste from the production process contains some amount of various additives, the additives are separated and removed in the pretreatment to increase the purity as follows. The scrap lead was first washed by deionized water and then its impurities (carbon powder and short fibre) were removed through precipitation, vacuum drying (at 80°C and 5 h), grinding (3 h) and sieving (less than 43 µm).

After the pretreatment, the remaining components are mainly lead sulfate, lead oxide and metal lead, their amount is more than 99 wt% by the chemical analysis. They are the raw material of 4BS synthesis.

Then, 4BS was prepared using the scrap lead paste obtained above, and the specific process is as follows.

Sulfuric acid with density of 1.25 g cm$^{-3}$ was weighed correctly and mixed with scrap lead paste of the corresponding amount in a dough-making machine (self-making ratio-speed 20 r.p.m.) to make five times the amount of the lead in the paste to sulfate ions (it comes from the stoichiometric ratio of $4PbO \cdot PbSO_4$), based on the quantitative analysis of the scrap lead paste after the pretreatment. It is left for 2 h and then diluted by deionized water to obtain pulp. The pulp is then stirred for 3 h at the rate of 1800 r.p.m. and 80°C in a magnetic stirrer. The pulp precipitate is then taken to a porcelain crucible and dried for 2 h at 80°C in a thermostat. The crucible is entered into a furnace and the

**Table 1.** Two mixing ways to prepare positive pastes.

| lead powder (kg) | 4BS (kg) | sulfuric acid (l) | deionized water (l) | fibre (g) |
| --- | --- | --- | --- | --- |
| 100 | 0 | 8.5 | 9–10 | 50 |
| 99 | 1 | 8.5 | 9–10 | 50 |

precipitate is pre-burnt for 1 h at 120°C and then is burnt at the temperature of 450°C and hold time of 7 h to synthesize 4BS with high purity and crystalline.

## 2.2. Characterization of 4BS

The crystalline phases of as-synthesized 4BS were characterized using X-ray diffraction (XRD). XRD analysis was carried out with a D8 ADVANCE diffractometer with Cu K$\alpha$ radiation ($\lambda = 1.5406$ Å), and the scan data were collected in the $2\theta$ range 10°–90° at scan rate 2° min$^{-1}$. The purity of 4BS was also briefly calculated with the XRD data and Materials Studio software according to a method provided by Lang *et al*. [9]. The morphology of the samples was observed by a scanning electron microscopy (SEM, JSM-6610A).

## 2.3. Electrochemical performance of VRLA batteries

Positive pastes were prepared in the conventional route. It was evenly mixed with additive lead powder, 4BS crystal seed (1 wt%), polypropylene fibre and graphite powder. And then deionized water and sulfuric acid ($\rho = 1.4$ g cm$^{-3}$) were added to the mixture being stirred. Two mixing ways to prepare positive pastes are presented in table 1. In this paper, the ratio of 4BS added to the positive paste was 0 wt% and 1 wt%. After the preparation of positive paste, the lead paste was pasted on the Pb–Ca–Sn–Al alloy grids and cured. The pasted Pb–Ca–Sn–Al alloy grids were cured for 12 h at 85°C and more than 95% humidity. The positive plates were made after drying for 36 h at 80°C–90°C and less than 20% humidity. The total curing time is 48 h. Positive plates were assembled with negative plates of commercial batteries to fabricate VRLA batteries (6-DZM-12) with absorptive glass mat (AGM) and the assembled batteries were formatted. The density of the electrolyte for formation was 1.25 g cm$^{-3}$ and the batteries underwent container formation (four charges and three discharges). Four batteries were fabricated and only two of them contain 4BS.

The VRLA batteries were measured on 2 h capacity. Charge accepting capacity, large current discharge capacity and low-temperature capacity at −18°C according to the international standard (DIN43539T2).

The VRLA batteries were simultaneously measured on 100% charge–discharge cycle life at 25°C. And then their charge–discharge curve was plotted. When the continued third-cycle capacity was 80% of the rating capacity, cycling was stopped and measured.

# 3. Results and discussion

## 3.1. Characterization of 4BS

4BS synthesized by heat treatment at 450°C and 7 h was ground up to less than 10 µm at normal room temperature, in the atmospheric environment and for 3 h by a ceramic ball-miller. The sample of 4BS was blackish yellow powder. XRD peaks of 4BS samples synthesized from lead scrap paste are shown in figure 2. As shown in figure 2, compared with the standard peaks (JCPDS No: 23-0333), the pattern corresponded to that of $Pb_5O_4SO_4$ very well and there were no impurity peaks in the XRD patterns. 4BS content in the sample was calculated by Material Studio software [9]. The calculation results are shown in figure 3.

The results showed that the content of 4BS is achieved *ca* 98.6%, and this means that the purity of the synthesized 4BS is very high. It demonstrated that 4BS can be effectively synthesized from scrap lead paste by the used method which is applicable to positive active material additive for improving the electrochemical performance and the cycle life of lead-acid batteries. SEM micrographs of the samples of synthesized 4BS from scrap lead paste for different magnifications are presented in figure 4. As

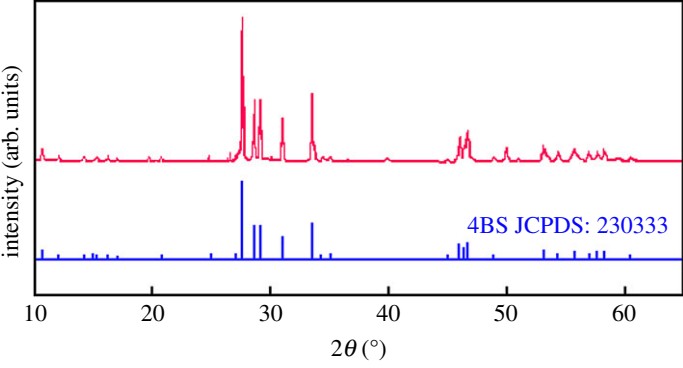

**Figure 2.** X-ray diffraction patterns for synthesized 4BS.

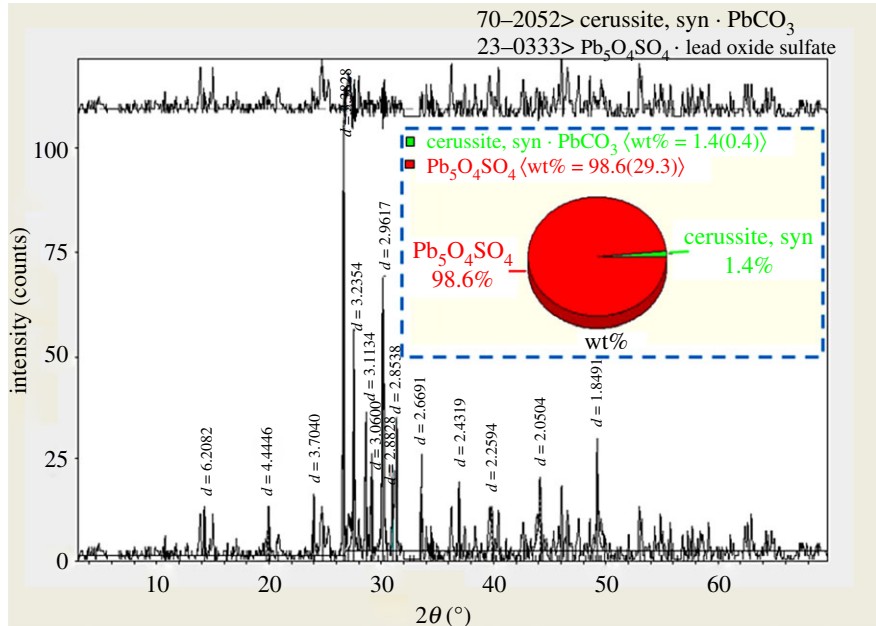

**Figure 3.** The content of synthesized 4BS in sample calculated by Material Studio software.

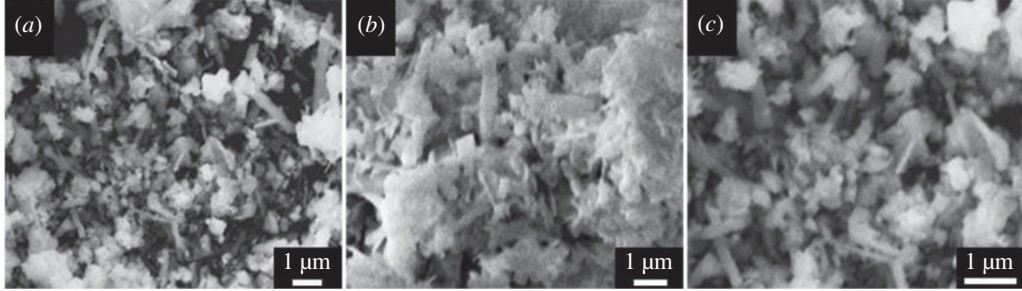

**Figure 4.** SEM images of 4BS in different magnification: (*a*) 10 000×, (*b*) 11 000× and (*c*) 18 000×.

shown in figure 4, the morphology of the particles of 4BS become more explicit and their size becomes bigger by increasing magnification. The size of particles is fine and some coagulants are also identified.

## 3.2. Phase characteristics of positive electrodes

### 3.2.1. Phase characteristics of plates after curing

The analysed result on the plate after curing is shown in table 2. In table 2, case 1 corresponds to the plate without the addition of 4BS, while case 2 is for the plate after adding 4BS as crystal seeds. As shown in

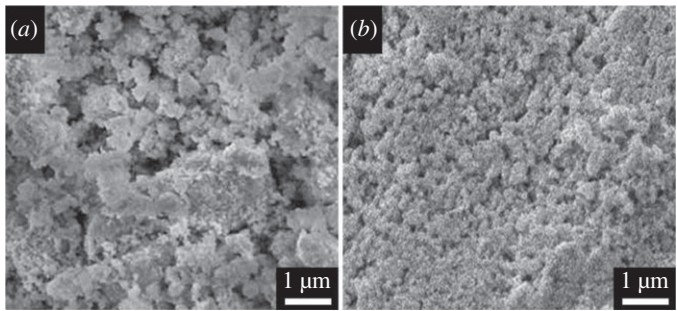

**Figure 5.** SEM images of plates before formation: (*a*) without the addition of 4BS and (*b*) after adding 4BS as crystal seeds.

**Table 2.** Composition of the plate after curing (wt%).

| battery number | $\omega$($\alpha$-PbO) | $\omega$($\beta$-PbO) | $\omega$(4BS) | $\omega$(3BS) | $\omega$(Pb) |
|---|---|---|---|---|---|
| 1 | 11.2 | 4.6 | 67.4 | 14.8 | 2.1 |
| 2 | 7.7 | 3.3 | 83.1 | 3.8 | 2.0 |

**Table 3.** Composition of the plate after formation (wt%).

| battery number | $\omega$($\alpha$-PbO$_2$) | $\omega$($\beta$-PbO$_2$) | $\omega$(PbSO$_4$) |
|---|---|---|---|
| 1 | 22.4 | 62.5 | 15.1 |
| 2 | 33.3 | 53.9 | 12.8 |

table 2, 4BS content of case 2 is 15.7 wt% higher than for case 1, whereas 3BS content is 11 wt% lower. The contents of $\alpha$-PbO and $\beta$-PbO of case 2 are, respectively, 3.5 wt% and 1.3 wt% lower than for case 1. From these results, we can see that adding 4BS as crystal seeds promotes the generation of 4BS in the lead paste during the curing process and it can be a type of ideal positive active material additive. For both cases, the Pb content is about 2 wt% that satisfies the requirements of the process. This means that adding 4BS as crystal seeds into positive active materials is probably efficient to increase the 4BS content in the plate after curing.

Figure 5 shows the morphology of positive active material without adding and with adding 4BS after curing. As clearly seen from figure 5*a*, the positive active material without adding 4BS is composed of the severely agglomerated rough particles. However, the positive active material with added 4BS as crystal seeds (figure 5*b*) has the homogeneous structure consisting of fine particles. This result indicated that the addition of 4BS to the positive active material facilitates the formation of the homogeneous structure with fine particles, which could enhance the performance of batteries.

### 3.2.2. Phase characteristics of plates after the formation

The analysed result on the plates after formation is given in table 3; case 2 corresponds to the composition of the plates without the addition of 4BS, while case 2 is for the plates after adding 4BS as crystal seeds.

As shown in table 3, although all the plates had undergone the same processes, the content of $\alpha$-PbO$_2$ in the plates after adding 4BS as crystal seeds is 10.9 wt% higher than that of the plate without 4BS and the content of $\beta$-PbO$_2$ is 8.6 wt% lower than that of the plates without the addition of 4BS. This shows that the addition of 4BS to plate leads to increase in the conversion rate of lead dioxide. As indicated in previous studies [8–10,16], $\alpha$-PbO$_2$ is advantageous for the performance of batteries, especially their cycle life, and adding 4BS as crystal seeds increases the content of $\alpha$-PbO$_2$ in the lead plates after formation.

## 3.3. Electrochemical performance of VRLA batteries

### 3.3.1. Initial performance measurement of VRLA batteries

Table 4 shows the measured result of the initial performance of the test batteries after adding 4BS as crystal seeds and the traditional batteries without the addition of 4BS. As shown in table 4, the initial

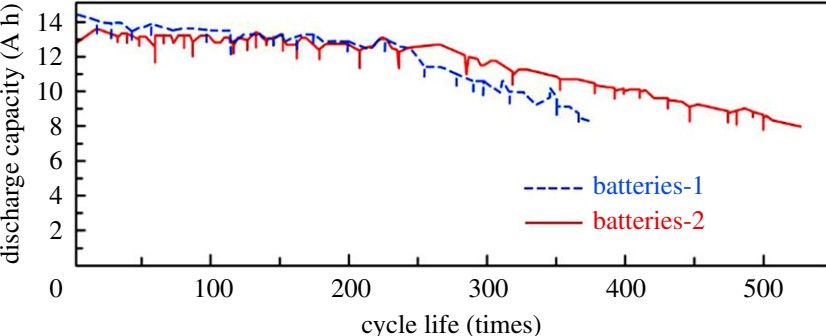

**Figure 6.** 100% charge–discharge cycle life.

**Table 4.** Initial performance of test batteries (A—without the addition of 4BSm, B—after adding 4BS as crystal seeds).

| battery number | initial discharge capacity (A h) | charge-accepting capacity | capacity at −18℃ (A h) | 1.8 C large current discharge capacity |
|---|---|---|---|---|
| A-1 | 13.1 | 1.857 | 10.9 | 27 min 37 s |
| A-2 | 13.2 | 1.795 | 10.8 | 26 min 58 s |
| B-1 | 12.7 | 2.078 | 11.0 | 28 min 42 s |
| B-2 | 12.9 | 2.103 | 11.2 | 29 min 23 s |

discharge capacity of the test batteries (B-1 and B-2) with 1 wt% 4BS added is lower about 2.3% than that of the traditional batteries (A-1 and A-2). It could be attributed to the easy formation of α-PbO$_2$ with rough particle by adding 4BS, and α-PbO$_2$ has lower specific capacity than β-PbO$_2$. However, the test batteries with 4BS added as crystal seeds exhibit the enhanced charge accepting capacity, −18℃ low-temperature capacity and 1.8 C large current discharge capacity. It means that for the performance of the test batteries, the initial discharge capacity was expected to be improved by adding 4BS as crystal seeds compared with the traditional batteries without the addition of 4BS. The improved electrochemical performance of batteries with 4BS could be attributed to the increase of the specific surface area that stemmed from the formation of a homogeneous structure with fine particles by adding 4BS.

### 3.3.2. Cycle performance measurement of VRLA batteries

Figure 6 shows the measurement result of the cyclic performance of test batteries after adding 4BS as crystal seeds and traditional battery without the addition of 4BS for comparison. As shown in figure 6, the traditional battery (battery-1) and the test battery (battery-2) have a significant difference of 100% charge–discharge cycle life. At the beginning of cycling, the test battery has obviously lower capacity than the traditional battery-1. After 120 times of cycling, all the batteries have similar capacity and after 250 times of cycling, the test battery was significantly superior to the traditional battery. 100% charge–discharge cycle life of the test battery was 523 times, while it was 365 times for the traditional battery. In addition, the test batteries with 4BS still kept their rating capacity. Therefore, the cycle life of the test batteries was about 1.4 times as long as the traditional batteries. Compared with the traditional battery, the cycle performance of the test battery with 4BS is more excellent. This means that the cycle life of positive active material with 4BS synthesized from scrap lead paste as crystal seeds are obviously improved.

In overall, when 4BS was added into positive active material of a VRLA battery as crystal seeds, a great amount of evenly distributed small particles of 4BS were formed on the plate after curing, as shown in table 2 and in figure 5. During the formation, a large amount of α-PbO$_2$ was produced due to 4BS (table 4).

The particle size of α-PbO$_2$ is much bigger than β-PbO$_2$ and it forms the skeleton structure, therefore α-PbO$_2$ is hardly softened and extracted during the charge and discharge cycle of a battery, and the cycle life of the battery is increased correspondingly. Therefore, 100% charge–discharge cycle life of the lead-acid batteries is greatly increased by adding 4BS of even 1 wt% as crystal seed. However, our study also showed that early softening and shedding were often in the run-out batteries. Future studies should aim

at the improvement of the synthesis method of 4BS, particle control of 4BS and variable content of 4BS in the positive paste.

# 4. Conclusion

In this work, progress has been made with regard to the increase of cycle life of lead-acid batteries. The synthesis and application of positive active material additive 4BS from the scrap lead paste was studied, which is one way to protect the environment and improve the electrochemical performance of lead-acid batteries.

The impurities were removed by the pretreatment of the scrap lead paste and then the result of 4BS content of 98.6 wt% was obtained by sintering for 7 h at 450°C. The samples of synthesized 4BS were reasonably pure and their particles were fine and evenly distributed by XRD and SEM test and Material Studio software calculation. Adding 1 wt% of synthesized 4BS as crystal seed, 2.3% of the initial discharge capacity of the battery was lower than original battery's, but its charge accepting capacity, −18°C capacity and 1.8 C large current discharge capacity were higher. And the 100% charge–discharge cycle life of the batteries with synthesized 4BS as crystal seeds (523 times) was about 1.4 times greater than that of the batteries without adding it (365 times). This means that the cycle life of positive active material with 4BS synthesized from scrap lead paste as crystal seeds are obviously improved.

Through the above research, we can know that adding 4BS synthesized from scrap lead paste as crystal seeds can improve the electrochemical performance of VRLA batteries effectively.

Data accessibility. All the available data for this work are presented within the paper.
Authors' contributions. My.K. participated in the experimental design, carried out the laboratory work and edited the manuscript. Mu.K. suggested the research topic and participated in data analysis. C.R. performed the electrochemical test. S.J. and I.P. performed XRD and SEM analyses. G.K. prepared all samples for analysis and interpreted the results. M.R. coordinated the study and helped draft the manuscript. All authors discussed results, helped edit the manuscript and gave final approval for publication.
Competing interests. We declare we have no competing interests.
Funding. This work was financially supported by the State Committee of Science and Technology, Democratic People's Republic of Korea.
Acknowledgements. We thank the State Committee of Science and Technology, Democratic People's Republic of Korea, for support.

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
