## [Reviewer comments · Royal Society Open Science]

Review History

RSOS-190176.R0 (Original submission)

Review form: Reviewer 1

Is the manuscript scientifically sound in its present form?

Yes

Are the interpretations and conclusions justified by the results?

Yes

Is the language acceptable?

Yes

Is it clear how to access all supporting data?

Not Applicable

Do you have any ethical concerns with this paper?

No

Have you any concerns about statistical analyses in this paper?

No

Recommendation?

Accept with minor revision (please list in comments)

Comments to the Author(s)

The authors proposed that report for tetrabasic lead sulfate was synthesized from scrap lead paste that was formed through the production process of the lead-acid batteries. The effectiveness lead-acid batteries adding of 4BS was evaluated, and the 100 % charge-discharge cycle life of the new battery (523 times) was about 1.4 times higher than that of general lead-acid batteries (365 times). The results are reasonable and the manuscript is well-organized. However, there are some minor problems about the manuscript, therefore I suggest this paper for "minor revision".

1# The schematic of fig 1 is not beautiful. The authors shall make it good.

2# In the introduction the authors may consider more energy storage material related references for broader readership. For example, DOI: (10.1039/C8CS00324F) (10.1002/AENM.201702093) (10.1021/ACSNANO.6B08069) (10.1007/S11705-018-1754-3) (10.1016/J.SYNTHMET.2017.10.014) (10.1007/S11705-018-1705-Z) (10.1021/ACSNANO.8B08079) (10.1002/ADOM.201700984) 10.1155/2019/2614327 ; DOI: 10.20964/2018.11.30; DOI: 10.20964/2017.09.06

3# The image quality of Fig 5 and Fig 6 are not good enough and suggest that the author modified them or re-capture to make good presentation. (Fig 5. Too small font size).

Review form: Reviewer 2

Is the manuscript scientifically sound in its present form?

No

Are the interpretations and conclusions justified by the results?

Yes

Is the language acceptable?

Yes

Is it clear how to access all supporting data?

Not Applicable

Do you have any ethical concerns with this paper?

No

Have you any concerns about statistical analyses in this paper?

No

Recommendation?

Reject

Comments to the Author(s)

This paper is regarding to the use of tetrabasic lead sulfate for positive materials additives in lead-acid battery use. As for the materials innovation and battery evaluation, the work is far less strong as publishable results. The materials characterization are limited, the battery evaluation need to be given as an average results as evaluated from multiple cells. SEM are blurred, TEM are

suggested to add and the XPS data are missing. The results and discussion need to be expanded and provide sound explanation.

Review form: Reviewer 3

Is the manuscript scientifically sound in its present form?

No

Are the interpretations and conclusions justified by the results?

No

Is the language acceptable?

Yes

Is it clear how to access all supporting data?

Yes

Do you have any ethical concerns with this paper?

No

Have you any concerns about statistical analyses in this paper?

Yes

Recommendation?

Major revision is needed (please make suggestions in comments)

Comments to the Author(s)

Kim, et al discussed their method of transforming scrap lead paste into useful 4BS for lead acid battery applications, which should be of interest for researchers related to this area. Following questions need to be addressed before considering publication:

1 experimental data needs to be provided to demonstrate the validity of figure 2 and figure 3;

2 in table 2, wt% (alfa-PbO₂) in battery 2 is less than that of battery 1, how come wt% (4BS) in battery 2 is more than that of battery 1?

3 Fig.4 is not clear to read;

4 Fig.5 cannot be used to distinguish different phases of PbO₂, BSD or other data should be given, also, the calculation process to get those weight ratios in table 2&3 should be outlined

5 the language in paper should be organized furthermore to present the work clearly and systematically

Decision letter (RSOS-190176.R0)

24-Apr-2019

Dear Mr Kim:

Manuscript ID: RSOS-190176

Title: "Study on synthesis and application of tetrabasic lead sulfate as the positive active material additive for lead-acid batteries"

Thank you for submitting the above manuscript to Royal Society Open Science. Your paper was sent to reviewers and their comments are included at the bottom of this letter.

In view of the concerns raised by the reviewers, the manuscript has been rejected in its current form. However, a new manuscript may be submitted which takes into consideration these comments.

Please note that resubmitting your manuscript does not guarantee eventual acceptance, and that your resubmission will be subject to peer review before a decision is made.

Your resubmitted manuscript should be submitted by 22-Oct-2019. If you are unable to submit by this date please contact the Editorial Office.

On behalf of the Subject Editor Professor Anthony Stace and the Associate Editor Professor Claire Carmalt

REVIEWER(S) REPORTS:
Associate Editor Comments to Author ():
RSC Associate Editor:
Comments to the Author:
(There are no comments.)

RSC Subject Editor:
Comments to the Author:
(There are no comments.)

Reviewers' Comments to Author:

Reviewer: 1

Comments to the Author(s)

The authors proposed that report for tetrabasic lead sulfate was synthesized from scrap lead paste that was formed through the production process of the lead-acid batteries. The effectiveness lead-acid batteries adding of 4BS was evaluated, and the 100 % charge-discharge cycle life of the new battery (523 times) was about 1.4 times higher than that of general lead-acid batteries (365 times). The results are reasonable and the manuscript is well-organized. However, there are some minor problems about the manuscript, therefore I suggest this paper for "minor revision".

1# The schematic of fig 1 is not beautiful. The authors shall make it good.

2# In the introduction the authors may consider more energy storage material related references for broader readership. For example, DOI: (10.1039/C8CS00324F) (10.1002/AENM.201702093) (10.1021/ACSNANO.6B08069) (10.1007/S11705-018-1754-3) (10.1016/J.SYNTHMET.2017.10.014) (10.1007/S11705-018-1705-Z) (10.1021/ACSNANO.8B08079) (10.1002/ADOM.201700984) 10.1155/2019/2614327; DOI: 10.20964/2018.11.30; DOI: 10.20964/2017.09.06

3# The image quality of Fig 5 and Fig 6 are not good enough and suggest that the author modified them or re-capture to make good presentation. (Fig 5. Too small font size).

Reviewer: 2

Comments to the Author(s)

This paper is regarding to the use of tetrabasic lead sulfate for positive materials additives in lead-acid battery use. As for the materials innovation and battery evaluation, the work is far less strong as publishable results. The materials characterization are limited, the battery evaluation need to be given as an average results as evaluated from multiple cells. SEM are blurred, TEM are suggested to add and the XPS data are missing. The results and discussion need to be expanded and provide sound explanation.

Reviewer: 3

Comments to the Author(s)

Kim, et al discussed their method of transforming scrap lead paste into useful 4BS for lead acid battery applications, which should be of interest for researchers related to this area. Following questions need to be addressed before considering publication:

1 experimental data needs to be provided to demonstrate the validity of figure 2 and figure 3;

2 in table 2, wt% (alfa-PbO₂) in battery 2 is less than that of battery 1, how come wt% (4BS) in battery 2 is more than that of battery 1?

3 Fig.4 is not clear to read;

4 Fig.5 cannot be used to distinguish different phases of PbO₂, BSD or other data should be given, also, the calculation process to get those weight ratios in table 2 should be outlined

5 the language in paper should be organized furthermore to present the work clearly and systematically

Author's Response to Decision Letter for (RSOS-190176.R0)

See Appendix A.

RSOS-190882.R0

Review form: Reviewer 1

Is the manuscript scientifically sound in its present form?

Yes

Are the interpretations and conclusions justified by the results?

Yes

Is the language acceptable?

Yes

Is it clear how to access all supporting data?

Yes

Do you have any ethical concerns with this paper?

No

Have you any concerns about statistical analyses in this paper?

No

Recommendation?

Accept with minor revision (please list in comments)

Comments to the Author(s)

After resubmission, the manuscript is more readable. Small comments: the figure 1 shall be of high pixel quality.

Review form: Reviewer 3

Is the manuscript scientifically sound in its present form?

No

Are the interpretations and conclusions justified by the results?

No

Is the language acceptable?

No

Is it clear how to access all supporting data?

Not Applicable

Do you have any ethical concerns with this paper?

No

Have you any concerns about statistical analyses in this paper?

Yes

Recommendation?

Major revision is needed (please make suggestions in comments)

Comments to the Author(s)

The authors should respond to the comments one by one and highlight the changes in the paper, moreover, I still don't see how the results in fig2 are obtained, supporting info on how and based on what the 4BS contents are calculated are needed, otherwise the data is questionable.

Decision letter (RSOS-190882.R0)

28-May-2019

Dear Mr Kim:

Title: Study on synthesis and application of tetrabasic lead sulfate as the positive active material additive for lead-acid batteries
Manuscript ID: RSOS-190882

The editor assigned to your paper has now received comments from reviewers. We would like you to revise your paper in accordance with the referee and Subject Editor suggestions which can be found below (not including confidential reports to the Editor). Please note this decision does not guarantee eventual acceptance.

Please submit a copy of your revised paper before 20-Jun-2019. Please note that the revision deadline will expire at 00.00am on this date. If we do not hear from you within this time then it will be assumed that the paper has been withdrawn. In exceptional circumstances, extensions may be possible if agreed with the Editorial Office in advance. We do not allow multiple rounds of revision so we urge you to make every effort to fully address all of the comments at this stage. If deemed necessary by the Editors, your manuscript will be sent back to one or more of the original reviewers for assessment. If the original reviewers are not available we may invite new reviewers.

RSC Associate Editor
Comments to the Author:
(There are no comments.)

Reviewers' Comments to Author:
Reviewer: 3

Comments to the Author(s)
The authors should responds to the comments one by one and highlight the changes in the paper, moreover, I still don't see how the results in fig2 are obtained, supporting info on how and based on what the 4BS contents are calculated are needed, otherwise the data is questionable.

Reviewer: 1

Comments to the Author(s)
After resubmission, the manuscript is more readable. Small comments: the figure 1 shall be of high pixel quality.

Author's Response to Decision Letter for (RSOS-190882.R0)

See Appendix B.

RSOS-190882.R1 (Revision)

Review form: Reviewer 1

Is the manuscript scientifically sound in its present form?

Yes

Are the interpretations and conclusions justified by the results?

Yes

Is the language acceptable?

Yes

Is it clear how to access all supporting data?

Not Applicable

Do you have any ethical concerns with this paper?

No

Have you any concerns about statistical analyses in this paper?

No

Recommendation?

Accept as is

Comments to the Author(s)

The authors have revised to the most of reviewers concerns.

Decision letter (RSOS-190882.R1)

11-Jun-2019

Dear Mr Kim:

Title: Study on synthesis and application of tetrabasic lead sulfate as the positive active material additive for lead-acid batteries

Manuscript ID: RSOS-190882.R1

It is a pleasure to accept your manuscript in its current form for publication in Royal Society Open Science. The chemistry content of Royal Society Open Science is published in collaboration with the Royal Society of Chemistry.

RSC Associate Editor:
Comments to the Author:
(There are no comments.)

RSC Subject Editor:
Comments to the Author:
(There are no comments.)

Reviewer(s)' Comments to Author:
Reviewer: 1

Comments to the Author(s)
The authors have revised to the most of reviewers concerns.

Appendix A

Dear Editor,

On behalf of my co-authors, we thank you very much for giving us an opportunity to revise our manuscript, we appreciate editor and reviewers very much for their positive and constructive comments and suggestions on our manuscript entitled "Study on synthesis and application of tetrabasic lead sulfate as the positive active material additive for lead-acid batteries" (ID: RSOS-190176). Those comments are all valuable and very helpful for revising and improving our paper, as well as the important guiding significance to our researches. We have studied the editor and reviewer's comments carefully. We have tried our best to revise our manuscript according to the editor and reviewer's comments. Attached please find the revised version, which we would like to submit for your kind consideration. Looking forward to hearing from you.

Thank you and best regards,

Yours sincerely,

Corresponding author:

Name: Myonghak Kim

E-mail: kmh311@163.com

The main corrections in the paper and the responds to the reviewer's comments are as flowing:

Reviewer: 1

Comments:

1. The schematic of Fig 1 is not beautiful. The authors shall make it good.

Respond to comment: Thanks for the referee's kind suggestion.

In the manuscript we submitted earlier, the schematic of Figure 1 was really not beautiful enough , so we made modification to the schematic of Figure 1.

2.In the introduction the authors may consider more energy storage material related references for broader readership.

Respond to comment: Thanks for the referee's kind suggestion

In the introduction, we supplemented some references related to energy storage materials for broader readership([27] and [28]).

3.The image quality of Fig 5 and Fig 6 are not good enough and suggest that the author modified them of re-capture to make good presentation.(Fig 5. Too small font size)

Respond to comment: Thanks for the referee's kind suggestion.

I think what I have done before is really not good enough. According to the reviewer's suggestion, Fig.4(Fig 5 on the originally submitted version of manuscript) and Fig 5(Fig 6 on the originally submitted version of manuscript) are modified in order to present the experimental results better.

Reviewer: 2

Comments:

This paper is regarding to the use of tetrabasic lead sulfate for positive materials additives in lead-acid battery use. As for the materials innovative and battery evaluation, the work is far less strong so publishable results. The materials characterization are limited, the battery evaluation need to be given as an average results as evaluated from multiple cells. SEM are blurred, TEM are suggested to add and the XPS data are missing. The results and discussion need to be expanded and provide sound explanation.

Respond to comment: Thank you for your valuable comments.

It is really true as Reviewer suggested that the materials characterization is limited. In fact, in this paper we have focused on comparing the electrochemical performance and the cycle life of lead-acid batteries that used the positive active material synthesized by adding 4BS and without 4BS. And by means of morphology analysis through SEM, it is verified that the addition of 4BS to the positive active material resulted in the refinement of particles and the homogenization of structure and it can be assumed that the improved performance is close related to the formation of the homogeneous structure with fine particles.

Review's valuable comments will become the important guiding to our researches. According to the reviewer's good instruction, we are able to

make a great effort to provide sufficient evidence for the materials characterization by means of effective analytical approach such as TEM and XPS.

Reviewer: 3

Comments:

1. Experimental data needs to be provided to demonstrate the validity of Figure 2 and Figure 3.

Respond to comment: Thank you for your valuable comments.

In order to demonstrate the validity of Figure 2 and Figure 3 (on the originally submitted version of manuscript), we made modification to the Figure 2 and give experimental data in order to better understand the results.

2. In table 2, wt%(α -PbO) in battery 2 is less than that battery 1, how come wt%(4BS) in battery 2 is more than that of battery 1?

Respond to comment: Thank you for your valuable comments.

In the process of curing, positive electrode material is transformed into tribasic lead sulfate (3BS) and tetrabasic lead sulfate (4BS) lead sulfate.

4BS is synthesized by the following reactions:

Or

As shown the above reaction, PbO plays an important role in the generation of 4BS. And adding 4BS as crystal seeds promote the generation of 4BS in the lead paste during the curing process. Therefore, adding 4BS as crystal seeds into positive active materials is efficient likely to increase the 4BS content in the plate after curing. Based on the above theory, we have reinterpreted this part.

3.Fig.4 is not clear to read?

Respond to comment: Thank you for your valuable comments.

Figure 3(Figure 4 on the originally submitted version of manuscript) has been modified.

4.Fig.5 cannot be used to distinguish different phases of PbO₂, BSD or other data should be giver, also, the calculation process to get those weight ratio in table 2 & 3 should be outlined.

Respond to comment: Thank you for your valuable comments.

XS. Lang et al.[8] was briefly calculated the purity of 4BS with the XRD data and Materials Studio software. According to a method provided by them, we also analyzed the purity of 4BS. The calculation results were shown in the Figure 4. The results shown that the content of 4BS is achieved ca. 98.6%, and this means that the purity of the synthesized 4BS is very high.

5.The language in paper should be organized furthermore to present the work clearly and systematically.

Respond to comment: Thank you for your valuable comments.

Referring to the suggestion of the reviewer, we tried our best to improve the manuscript and made some changes in the manuscript. These changes will not influence the content and framework of the paper.

We appreciate for Editors/Reviewers' warm work earnestly, and hope that the correction will meet with approval.

Once again, thank you very much for your comments and suggestions.

Appendix B

Dear Editor,

On behalf of my co-authors, we thank you very much for giving us an opportunity to revise our manuscript, we appreciate editor and reviewers very much for their positive and constructive comments and suggestions on our manuscript entitled "Study on synthesis and application of tetrabasic lead sulfate as the positive active material additive for lead-acid batteries" (ID: RSOS-190882). Those comments are all valuable and very helpful for revising and improving our paper, as well as the important guiding significance to our researches. We have studied the editor and reviewer's comments carefully, and have made revision which marked in red in the revised manuscript. We have tried our best to revise our manuscript according to the editor and reviewer's comments. Attached please find the revised version, which we would like to submit for your kind consideration. Looking forward to hearing from you.

Thank you and best regards,

Yours sincerely,

Corresponding author:

Name: Myonghak Kim

E-mail: kmh311@163.com

The main corrections in the paper and the responds to the reviewer's comments are as flowing:

Reviewer: 1

Comments:

After resubmission, the manuscript is more readable. Small comments:
the Figure 1 shall be of high pixel quality.

Respond to comment: Thanks for the referee's kind suggestion.

As the reviewer mentioned, the schematic of Figure 1 was really not beautiful enough. In order to improve the pixels of the schematic of Figure 1, we made modification to the schematic of Figure1.

Reviewer: 3

Comments:

The authors should responds to the comments one by one and highlight the changes in the paper, moreover, I still don't see show the results in fig 2 are obtained, supporting info on how and based on what the 4BS contents are calculated are needed, otherwise the data is questionable.

Respond to comment: Thank you for your valuable comments.

As the reviewer mentioned, we are not giving sufficient evidence for the results of Figure 2. So we removed this part (including Figure 2), and we guarantee that this will not affect the overall content of the paper.

In our paper, We calculated the content of 4BS synthesized by sintering at temperature of 450 °C and hold time of 7 h using the XRD data and Materials Studio software, the results are shown in Figure 2 and Figure 3

of the revised version.

We tried our best to improve the manuscript and made some changes in the manuscript. These changes will not influence the content and framework of the paper. And here we did not list the changes but marked in red in revised paper.

We appreciate for Editors/Reviewers' warm work earnestly, and hope that the correction will meet with approval.

Once again, thank you very much for your comments and suggestions.